# Key Factors in Enhancing Pseudocapacitive Properties of PANI-InO_x_ Hybrid Thin Films Prepared by Sequential Infiltration Synthesis

**DOI:** 10.3390/polym15122616

**Published:** 2023-06-08

**Authors:** Jiwoong Ham, Hyeong-U Kim, Nari Jeon

**Affiliations:** 1Department of Materials Science and Engineering, Chungnam National University, Daejeon 34134, Republic of Korea; maxbasic55@o.cnu.ac.kr; 2Department of Plasma Engineering, Korea Institute of Machinery & Materials (KIMM), Daejeon 34103, Republic of Korea; guddn418@kimm.re.kr

**Keywords:** indium oxide, polyaniline, cyclic voltammetry, sequential infiltration synthesis (SIS), conducting polymer

## Abstract

Sequential infiltration synthesis (SIS) is an emerging vapor-phase synthetic route for the preparation of organic–inorganic composites. Previously, we investigated the potential of polyaniline (PANI)-InO_x_ composite thin films prepared using SIS for application in electrochemical energy storage. In this study, we investigated the effects of the number of InO_x_ SIS cycles on the chemical and electrochemical properties of PANI-InO_x_ thin films via combined characterization using X-ray photoelectron spectroscopy, ultraviolet–visible spectroscopy, Raman spectroscopy, Fourier transform infrared spectroscopy, and cyclic voltammetry. The area-specific capacitance values of PANI-InO_x_ samples prepared with 10, 20, 50, and 100 SIS cycles were 1.1, 0.8, 1.4, and 0.96 mF/cm², respectively. Our result shows that the formation of an enlarged PANI-InO_x_ mixed region directly exposed to the electrolyte is key to enhancing the pseudocapacitive properties of the composite films.

## 1. Introduction

Organic–inorganic hybrid materials have received continued attention owing to their novel functionalities, which are not demonstrated in single-phase materials, whether organic or inorganic. In particular, electrically conductive organic–inorganic composite films have demonstrated enhanced properties for application in electrochemistry-related areas. Some recently reported examples of the benefits of using PANI–metal oxide composite films include enhanced efficiency in photovoltaic cells [1], sensitivities and linearities in sensors [2], photocatalytic efficiencies in catalysts [3], and retention properties in supercapacitors [4]. The different properties of hybrid thin films and their performances in different areas depend primarily on the synthesis routes of the thin films. Therefore, the development of novel techniques for preparing organic–inorganic hybrid thin films has received attention in various areas.

Sequential infiltration synthesis (SIS) is a new vacuum-based technique for preparing organic–inorganic composites and is considered a variant of atomic layer deposition (ALD). Although ALD exploits self-limiting chemical reactions on the surfaces, SIS utilizes chemical reactions within the bulk polymer phase [5]. For SIS reactions to occur readily, the precursors designated to be used must first infiltrate the polymer phases. Typically, the entrapment of infiltrated SIS precursors by specific functional groups of the polymer is preferred. Poly(methylmethacrylate) (PMMA) [6] and poly(2-vinylpyridine) (P2VP) [7] are typical types of polymers that have been widely used for SIS because the carbonyl groups in PMMA and pyridine groups in P2VP undergo Lewis acid–base reactions with typical SIS precursors such as trimethylaluminum and titanium tetrachloride (TiCl_4_). However, the potential applications of hybrid thin films based on PMMA and P2VP are limited to areas where electrical conductivity is not required. Only several studies have been conducted on SIS with conducting polymers and their applications in electrochemistry.

Polyaniline (PANI) is a representative conducting polymer with controllable electrical conductivity owing to doping. PANI can exist in three different chemical states, namely, leucoemeraldine, pernigraniline, and emeraldine, depending on the degrees of oxidation and reduction [8,9]. PANI with an emeraldine base can be transformed into an emeraldine salt, which exhibits high electrical conductivity (~10^2^ S/cm) when acid-doped [10]. Wang et al. reported the SIS process of doping PANI with SnCl_4_ and MoCl_5_ vapors [11], which exhibit the Lewis acidic nature; the doped PANI exhibited a moderate conductivity of ~9.8 × 10^−5^ S/cm. PANI-ZnO (~18.42 S/cm) [12] and PANI-InO_x_ (4–9 S/cm) [4] composite thin films prepared via SIS showed electrically conductive properties. Previously, we demonstrated the significant potential of PANI-InO_x_ composite films prepared using SIS for electrochemical energy storage, which warrants follow-up studies on the same system [4].

The aim of this study was to investigate the influence of metal oxides on the electrochemical properties of polyaniline–metal oxide composites as energy storage materials utilizing the SIS. Research related to SIS focusing on conducting polymers is limited to several papers, and studies specifically investigating their electrochemical properties are scarce. In this study, we investigated the variations in the chemical and electrochemical properties of PANI-InO_x_ films prepared via SIS as a function of the SIS cycle number. PANI-InO_x_ films exhibit a graded concentration of InO_x_ along the direction of the film thickness, where their structure is determined by the number of cycles. A combination of ultraviolet–visible (UV-vis) spectroscopy, Raman spectroscopy, and attenuated total reflectance–Fourier transform infrared (ATR-FTIR) spectroscopy was performed to understand the variation in the chemical structure of PANI in response to alloying with InO_x_. The superior pseudocapacitive properties of the sample with the optimized cycle number (50 cy) are attributable to the increased volume of the PANI-InO_x_ mixed region, which is exposed to the electrolyte.

## 2. Materials and Methods

### 2.1. Sample Preparation

PANI with an emeraldine base powder (M_w_ ~10,000, Sigma–Aldrich, Saint Louis, MI, USA) was dissolved in methyl-2-pyrrolidone (≥99%, Sigma–Aldrich, Saint Louis, MI, USA) with a concentration of 30 mg/mL. The solution was stirred for 24 h at 80 °C and 850 rpm. The solution was spun onto prepared substrates at 2000 rpm, and the as-spun substrates were baked at 70 °C in air. The thickness of the prepared PANI thin films was approximately 37 nm. Subsequently, SIS was performed using a thermal ALD reactor (Daeki HighTech, Daejeon, Republic of Korea) with a cross-flow design. The precursors used for the SIS were trimethylindium (TMIn, 99.999%, EasyChem) and H_2_O (99.999%, Sigma–Aldrich, Saint Louis, MI, USA). Ar carrier gas (99.999%) was continuously flowed at 5 sccm during the entire SIS. Both the TMIn and H_2_O half-cycles involved a 1 s dose, 120 s of exposure, and 120 s of purging. The reactor chamber was isolated from the pump during the exposure step to facilitate the infiltration of the precursors into the polymer matrix. The SIS-prepared substrates were annealed at 270 °C for 1 h in the forming gas of H_2_–N_2_ (~3.9% H_2_ in N_2_).

Different types of substrates were used for different characterization methods: an Si substrate (n type, 1–10 Ohm·cm, iTASCO) with a 500-nm-thick SiO_2_ layer was used for X-ray photoelectron spectroscopy (XPS) and Raman spectroscopy. Au-coated Si substrate (Au thickness: ~90 nm) was used for ATR-FTIR spectroscopy. A fused silica (iNexus, Inc., Seongnam, Republic of Korea), which had a transmittance of ~90% or higher in the wavelength range > 250 nm, was utilized for UV-vis spectroscopy. Electrochemical experiments were performed using glass substrates with a fluorine-doped tin oxide (FTO) layer measuring ~600 nm thick (NSG TEC 7, Pilkington, Lathom, UK).

### 2.2. Sample Characterization

HRXPS depth profiling was performed using an X-ray photoelectron spectrometer (K-alpha, Thermo Scientific, Waltham, MA, USA) with Ar^+^ ion beams at 1 kV and an etching time of 10 s. The X-ray source used was monochromatic Al Kα (1487 eV). The surfaces of the annealed samples were partly scratched using stainless-steel tweezers to create a surface step on the sample, and the thickness of the PANI-InO_x_ thin film was measured using a stylus profiler (Alpha-Step^®^ D-500, KLA, Milpitas, CA, USA). The cleanliness of the scratched surface was confirmed based on spectroscopic ellipsometry data (FS-1, Film Sense, Lincoln, NE, USA) obtained from the surface and a comparison with those of a bare Si substrate. Raman spectroscopy was performed using a Raman spectrometer (LabRAM HR-800, Horiba, Japan) under the following conditions: 514 nm laser source measuring 0.7 μm, 1800 gr/mm grating, 10 s acquisition time, and 10 specular accumulations. UV-vis spectroscopy was performed using a UV-vis spectrometer (UV-2600, Shimadzu, Japan), where an FTIR spectrometer (Vertex 80v, Bruker, Billerica, MA, USA) with a mercury–cadmium–telluride detector and a diamond attenuated total reflection (ATR) crystal were used to obtain the ATR-FTIR spectroscopy data at a spectral resolution of 4 cm^−1^. An electrochemical analyzer (CHI602E, CH Instruments, Bee Cave, TX, USA) was used to perform cyclic voltammetry (CV) experiments using a three-electrode setup comprising a Ag/AgCl reference electrode, a Pt wire counter electrode, and a PANI-InO_x_ FTO/glass working electrode. CV data were obtained using a pH 7 buffer solution as the electrolyte.

## 3. Results and Discussion

We analyzed PANI-InO_x_ composite thin films, which were prepared under different SIS cycles (10, 20, 50, and 100 cy) and annealed in a reducing atmosphere. The sample structure could be summarized as follows: (1) an InO_x_-rich region, (2) a PANI-InO_x_ mixed region, and (3) a PANI-rich region, as shown in Figure 1a. The thicknesses and chemical compositions of the three regions differed depending on the number of SIS cycles (Figure 1b). The samples with 10 and 20 SIS cycles did not contain an InO_x_ surface layer and only presented a PANI-InO_x_ mixed region and a PANI bulk region. Owing to the repetition of the SIS cycle, the InO_x_-rich region near the surface of the PANI at times prevented the additional infiltration of the TMIn precursor, thereby resulting in the formation of a thicker InO_x_ layer, which further developed via a mechanism similar to ALD. However, TMIn infiltration in later SIS cycles may not have been completely hindered because the concentration in the PANI-rich region of the 100 cy sample decreased gradually from ~40 to 0 at%.

Figure 1c shows the oxygen (O) and nitrogen (N) HRXPS data obtained at different locations on the PANI-InO_x_ film, as shown in Figure 1b. The O 1s HRXPS data were deconvoluted into three or four peaks originating from the following components: lattice oxygen (In-O) from InO_x_ at ~529.9 eV, oxygen vacancy ($V_{o}^{˙˙}$) at ~531.0 eV, indium hydroxide (In-OH) at ~532.1 eV, and lattice oxygen (Si-O) from SiO_2_ at ~533.1 eV [13,14]. In all four samples, the In-O component was the most dominant in the topmost region, which was either an InO_x_-PANI mixed region (samples with 10 and 20 SIS cy) or InO_x_-rich regions (Samples with 50 and 100 SIS cy). The components of higher BEs (i.e., −OH and $V_{o}^{˙˙}$) became more dominant compared to the In-O component as the HRXPS analysis region shifted toward the substrate (i.e., PANI-rich region). This is consistent with the previous SIS results, which indicated that the oxidation of the SIS precursors within the polymer matrix was less complete than that on the polymer surface [6,15]. The average stoichiometries of the four samples were InO_0.85_, InO, InO_1.38_ and InO_1.44_ for 10, 20, 50, and 100 cycles, respectively. The stoichiometry trend was reasonable, considering that the proportion of surface-like InO_x_ compared with that of bulk-like InO_x_ (i.e., synthesized within the polymer matrix) enhanced as the SIS cycle number increased.

The N 1s HRXPS data of the four samples were deconvoluted into three components: quinonoid imine (−N=) at ~398.4 eV, benzenoid amine (−NH−) at ~399.5 eV, and protonated amine/imine state (−NH_2_^+^–, =NH^+^−) at ~400.4 eV [16]. In all the HRXPS spectra, the amine component was more dominant than those of the other components. Meanwhile, the PANI with an emeraldine base usually contained equal amounts of imine and amine components. The presence of protonated species along with a decrease in the number of imine units suggested that the protonated species may have originated from the imine units. PANI doped with HCl contains protonated species transformed from the imine components [17]. However, no clear correlation was indicated between the percentage of protonated species and the InO_x_ content in any sample. Therefore, further studies are necessary to determine the potential chemical reactions contributing to the formation of protonated species during InO_x_ alloying.

Figure 2a shows the UV-vis transmittance spectra of the four samples. The samples with 10, 20, and 50 cy showed a weak absorption band at ~610 nm, which was assigned to n–π* transition between the benzenoid and quinonoid rings. The increase in the absorption below ~400 nm from the three sample was likely related to the π–π* transition of the benzenoid ring, which is known to be located at ~320 nm [18]. The 100 cy sample showed stronger absorption in the UV region (<400 nm), whereas the n–π* transition was primarily suppressed. The absorption in the UV region was likely due to absorption by the thick InO_x_ layer. The Tauc plot of the 100 cy SIS shows that the bandgap of InO_x_ was ~3.5 eV, which was slightly lower level than that of the dense InO_x_ thin films reported in the literature [19,20]. The smaller bandgap of InO_x_ along with the presence of tail states was reasonable, considering that a significant portion of the InO_x_ phase was present within the polymer matrix along with a high concentration of oxygen vacancies. The optical bandgap of PANI-InO_x_ samples tends to decrease as the number of SIS cycles decreases (Appendix A). The ATR-FTIR spectra of the four samples (Figure 2b) showed IR bands related to the PANI phase: (1) stretching of the quinonoid ring at ~1600 cm^−1^, (2) stretching of the benzenoid ring at ~1512 cm^−1^, (3) stretching of C–N of the secondary aromatic amine at 1300 cm^−1^, and (4) out-of-plane C–H deformation of the 1,4-distributed aromatic ring at 823 cm^−1^ [21,22,23]. The 100 cy samples showed significant IR bands, with lower intensities compared with the other samples owing to the presence of the InO_x_ surface layer. Similarly, the Raman spectra of the four samples (Figure 2c) exhibited significant Raman bands associated with the PANI phase, as summarized in Table 1. The presence of the C–N^+•^ stretching (radical cations) band at ~1350 cm^−1^ in all samples is consistent with the presence of protonated amine/imine species indicated in the HRXPS analysis. Radical cation bands are typically observed in acid-doped PANI, which suggests that alloying with InO_x_ may offer similar effects on the acid doping of PANI.

The CV results for the samples, measured at a scan rate of 10 mV/s, are shown in Figure 3. All the samples showed a pair of redox peaks at similar potentials (i.e., ~0.2 V vs. Ag/AgCl and ~−0.05 V vs. Ag/AgCl). The 50 cy sample indicated better-defined redox peaks with a higher current compared with the other samples. The area-specific capacitance values of PANI-InO_x_ samples prepared with 10, 20, 50, and 100 SIS cycles were 1.1, 0.8, 1.4, and 0.96 mF/cm², respectively. A detailed explanation of this calculation is reported in the literature [4]. The CV measurements were performed multiple times using different samples prepared under the same SIS conditions. The redox peak positions of the CV curves varied slightly within a ~100 mV range. Therefore, the subtle variation in the peak position observed for the different SIS samples (10, 20, 50, and 100 cy) is considered to be within the experimental error. CV curves collected at different scan rates are provided in Appendix A. In order to investigate the capacitance stability of the sample after prolonged exposure to the electrolyte, we conducted a CV experiment consisting of 1000 cycles (Appendix A). This experiment assessed the evolution of the capacitance over time in response to extended electrolyte exposure. In the electrochemical impedance spectroscopy (EIS) test conducted on the PANI-InO_x_ 50 cycle, pure PANI, and pure InO_x_ samples in a previous study, semicircles were observed at high frequencies and straight lines were observed at low frequencies [4]. Among the three samples, the composite sample exhibited the smallest semicircle, indicating a lower charge transfer resistance.

Further analysis is necessary to identify the redox reactions contributing to the observed CV peaks. However, the conversion of the emeraldine and pernigraniline states was speculated to be the primary redox reaction in the PANI-InO_x_ samples in our previous study. The enhanced peak current of the 50 cy sample might have been related to the larger thickness of the PANI-InO_x_ mixed region (Figure 1b) compared with those of the 10 and 20 cy samples. The 100 cy sample exhibited a sufficiently thick PANI-InO_x_ mixed region; however, the InO_x_ surface layer likely prevented direct contact between the mixed region and the electrolyte [4]. Furthermore, the 50 cy sample had a larger proportion of protonated amine/imine structures (Figure 1c), a fact which is likely related to the PANI-InO_x_ mixed region.

## 4. Conclusions

It is only recently that SIS was proved to be a promising route for the preparation of organic–inorganic hybrid films for electrochemistry and electrochemical energy storage. The goal of this study was to provide a more comprehensive understanding of the effects of the number of SIS cycles (10, 20, 50, and 100 cycles) on the chemical and electrochemical properties of PANI-InO_x_ composite films. The PANI-InO_x_ films showed a graded composition of InO_x_ within the PANI matrix, with ample concentrations of oxygen vacancies and hydroxide components. The entire film structure was composed of two or three components, including an InO_x_-rich region, a PANI-InO_x_ mixed region, and a PANI-rich region, and its structure depended on the number of cycles. Combined characterization using HRXPS, UV-vis, Raman, and FTIR spectroscopy consistently revealed the presence of cationic radicals, which might have been related to the transition from the quinonoid structure to the benzenoid structure. The 50 cy samples showed the highest pseudocapacitance among the tested samples, which was likely due to the relatively thick electrochemically active PANI-InO_x_ region and the exposure of the PANI-InO_x_ region to the electrolyte.

## Figures and Tables

**Figure 1 polymers-15-02616-f001:**
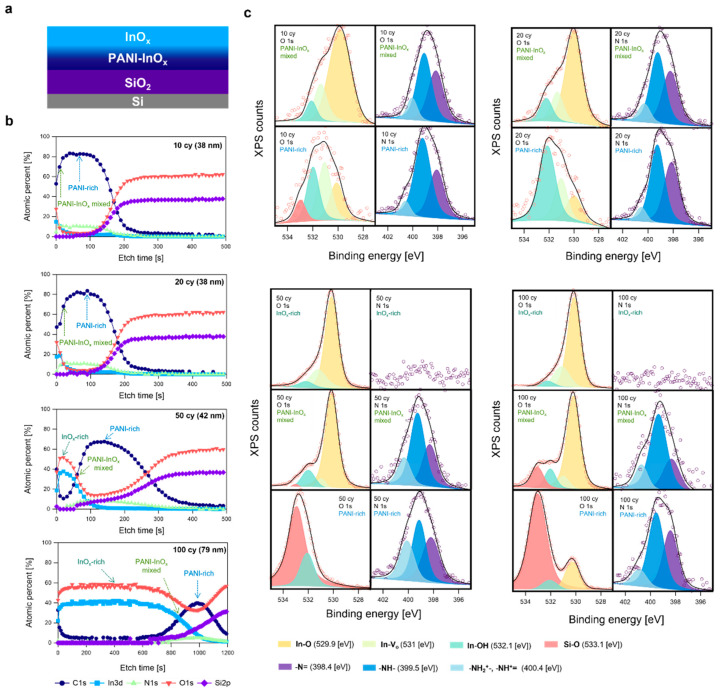
(**a**) Schematic illustration of structure of PANI-InO_x_/SiO_2_/Si samples. InO_x_ content shows graded concentration along film thickness direction. (**b**) XPS depth profiles showing C, In, N, O, and Si atomic concentrations for four SIS samples with different cycle numbers: 10, 20, 50, and 100 cy. (**c**) O 1s and N 1s HRXPS data obtained at different locations (i.e., InO_x_-rich region, PANI-InO_x_ mixed region, and PANI-rich region) in four samples. Each location at which HRXPS data were captured are shown as arrows in (**b**).

**Figure 2 polymers-15-02616-f002:**
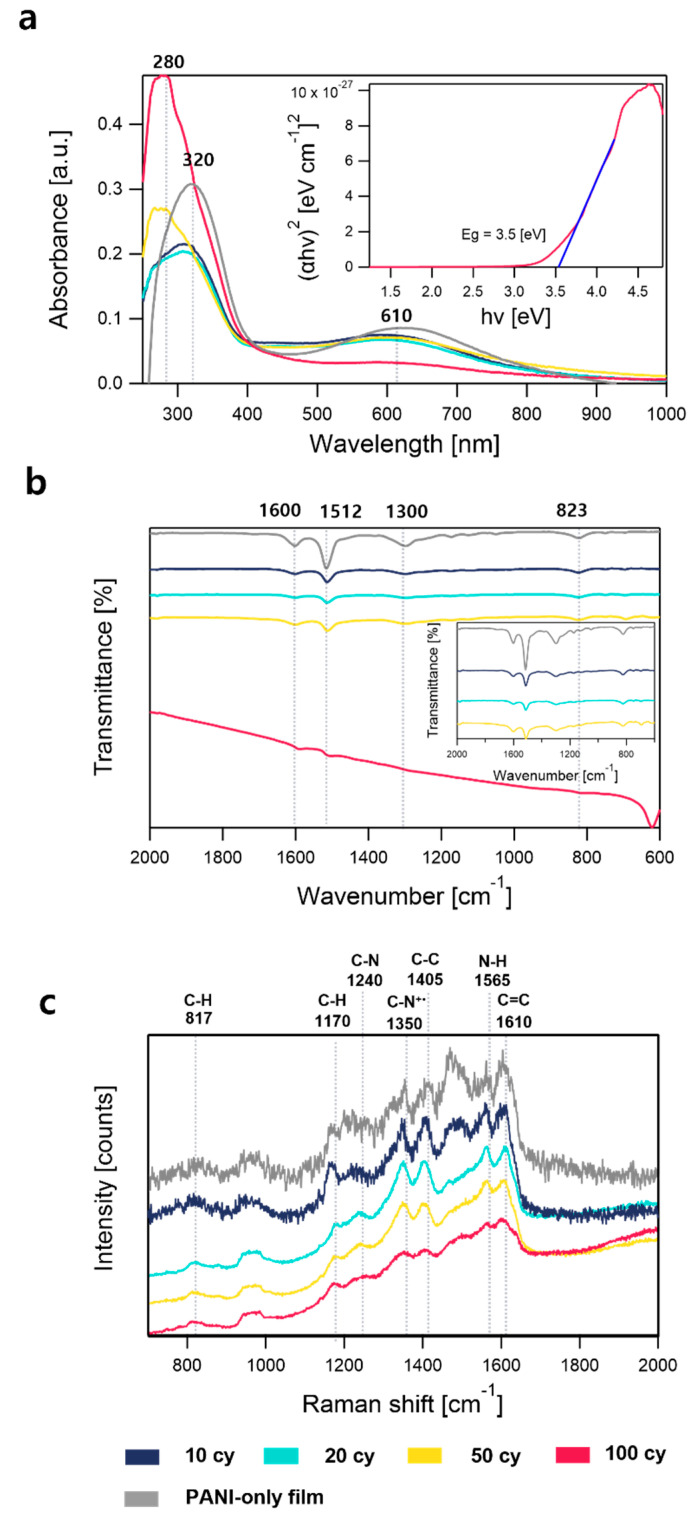
(**a**) UV-vis transmittance spectra, (**b**) ATR-FTIR absorbance spectra, and (**c**) Raman spectra of PANI-InO_x_ samples at different number of cycles (10, 20, 50, and 100 cy) and PANI-only sample. The PANI-only sample was annealed under the same conditions as the PANI-InO_x_ samples. Inset of (**a**) shows Tauc plot of 100 cy PANI-InO_x_ sample. The dashed lines in panel (**c**) mark the location of Raman bands observed in the PANI-InO_x_ film, which are also summarized in Table 1.

**Figure 3 polymers-15-02616-f003:**
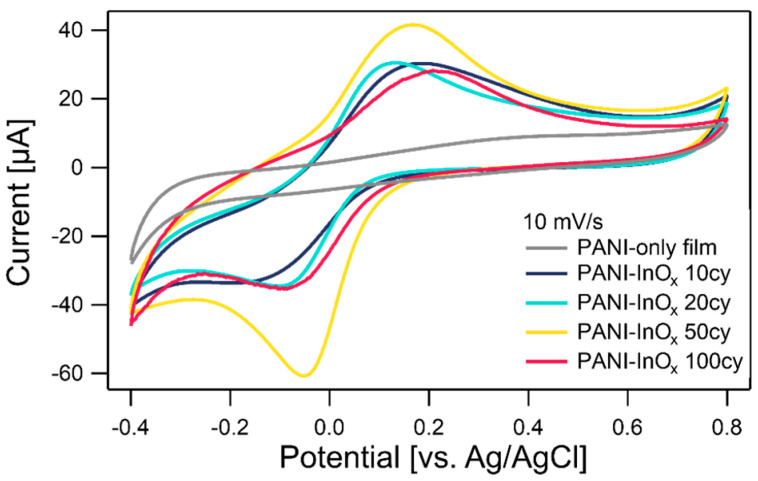
CV curves of PANI-InO_x_ films prepared with different SIS cycles (10, 20, 50, and 100 cy) and PANI-only film. The CVs were collected at a scan rate of 10 mV/s in an aqueous electrolyte of neutral pH.

**Table 1 polymers-15-02616-t001:** Major Raman bands identified in the PANI-InO_x_ and PANI-only samples.

Raman Shift (cm^−1^)	Assignment
1610	C=C stretching vibration of a quinonoid ring
1560	N–H bending
1405	C–C stretching vibrations in a quinonoid ring
1350	C–N^+•^ Radical cation
1240	C–N stretching in a benzenoid ring
1170	C–H in-plane C–H bending quinonoid ring
817	out-of-plane C–H vibration

## Data Availability

The data presented in this study are available in the article.

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
