# Peer review of "Key Factors in Enhancing Pseudocapacitive Properties of PANI-InOx Hybrid Thin Films Prepared by Sequential Infiltration Synthesis"

_polymers, 2023, doi:10.3390/polym15122616_

Round 1

Reviewer 1 Report

The authors have provided very detailed experimental details on characterization and film property analysis; however, they should provide more data on electrochemical performance. Especially, they are recommended to study effect of long-time exposure of these films to electrolyte and how it impacts dissolution of these films into electrolyte. They can do a time-based study where they can collect CVs at different timestamp to study impact of electrolyte interaction with these films to further support their claims on pseudo-capacitance. Author is also recommended to collect CV on only PANI film. Along with CV plot, in order to claim impact on pseudo-capacitance, author should characterize capacitance with regards to film mass (or thickness).

Reviewer 2 Report

In this manuscript, the authors reported "Key Factors in Enhancing Pseudocapacitive Properties of PANI-InOx Hybrid Thin Films Prepared by Sequential Infiltration Synthesis". The work is interesting. I recommend it be accepted for publication after moderate revision. The main concerns are as follows.

1.      In the abstract: The authors need to include the numerical results of the pseudocapacitive properties of the composite thin films.

2.      In the introduction: The authors should include the recent literature survey of the PANI-metal oxide composite thin films for electrochemical characteristics.

3.      The authors should give justification to prove the novelty of this research.

4.      The authors need to check the caption of Table 1.

5.      In Figure 2: The authors need to include the optical bandgap (Tauc's plot) of PANI-InOx (10, 20, and 50 cy) composite thin films.

6.      In Figure 2a: Check the x-axis wavenumber [nm] or wavelength [nm]??? Check it seriously.

7.      In Figure 2b: Check the y-axis unit.

8.      In Figure 3: The authors add the CV analysis of the prepared composite thin films with respect to different scan rates.

9.      The authors need to include the EIS analysis of the prepared composite thin film, which should be discussed briefly.  

10.  The novelty should be highlighted in the conclusion.

11.  Improve the figure captions and add more details.

12.  English expressions should be improved.

English expression should be improved. 

Round 2

Reviewer 1 Report

Authors have taken into account all the revisions I had recommended and provided additional data on long exposure to electrolyte. They also updated capacitance values based on area of electrode, which now makes the results strongly conclusive.